# Biomechanical Analysis of Recreational Cycling with Unilateral Transtibial Prostheses

**Heloísa Seratiuk Flores** [1,*], **Wen Liang Yeoh** [2], **Ping Yeap Loh** [3], **Kosuke Morinaga** [4] and **Satoshi Muraki** [3]

1   Graduate School of Design, Kyushu University, Fukuoka 815-8540, Japan
2   Faculty of Science and Engineering, Saga University, Saga 840-8502, Japan; wlyeoh@cc.saga-u.ac.jp
3   Faculty of Design, Kyushu University, Fukuoka 815-8540, Japan; py-loh@design.kyushu-u.ac.jp (P.Y.L.); muraki@design.kyushu-u.ac.jp (S.M.)
4   Faculty of Rehabilitation, Hiroshima International University, Higashihiroshima 739-2631, Japan; morinaga@hirokoku-u.ac.jp
*   Correspondence: hflores@kyudai.jp

**Abstract:** Leg prostheses specially adapted for cycling in patients with transtibial amputation can be advantageous for recreational practice; however, their required features are not fully understood. Therefore, we aimed to evaluate the efficiency of unilateral cycling with a transtibial prosthesis and the characteristics of different attachment positions (middle and tip of the foot) between the prosthetic foot and the pedal. The cycling practice was performed on an ergometer at 40 W and 60 W resistance levels while participants (n = 8) wore custom-made orthoses to simulate prosthesis conditions. Using surface electromyogram, motion tracking, and power meter pedals, biomechanical data were evaluated and compared with data obtained through regular cycling. The results showed that power delivery became more asymmetrical at lower workloads for both orthosis conditions, while hip flexion and muscle activity of the knee extensor muscles in the sound leg increased. While both pedal attachment positions showed altered hip and knee joint angles for the leg wearing the orthosis, the middle of the foot attachment presented more symmetric power delivery. In conclusion, the middle of the foot attachment position presented better symmetry between the intact and amputated limbs during cycling performed for rehabilitation or recreation.

**Keywords:** kinematics; electromyography; unilateral amputee cycling; recreational cycling; prosthesis design; human factors design

## 1. Introduction

Any type of physical exercise can positively affect the rehabilitation of patients with lower limb amputations, by improving the psychological and physical conditions of those who are engaged in it. The psychological factor plays an important part in regaining independence and adapting to the use of prostheses [1]. Cycling is a low-impact exercise that provides the benefits of physical exercise to patients with amputations who are performing it at a recreational level and aids in familiarization with the use of prostheses, possibly serving as a rehabilitation tool [2–7]. During cycling, the body's weight is supported independent of the motion or support of the legs, making it simpler to move them. Furthermore, when cycling on an ergometer or a stationary bike, balancing becomes easier. However, amputee cycling is often studied at the professional level only, and its potential for rehabilitation and recreational applications is only now being discovered.

A recent study [8] investigated the cycling habits of individuals with lower limb amputation in the Netherlands. Of the 207 participants, 141 (68%) practiced cycling after lower limb amputation, with 80% practicing it for recreational purposes and 74% for physical fitness. Although most participants used daily walking prostheses (33%, with 4% wearing cycling prostheses), 19.1% stated that they experienced pain or discomfort while cycling, and 6.4% specifically stated that they had problems cycling with the daily

prostheses. These data indicate the high popularity of cycling among patients with lower limb amputation as an exercise, as well as possible issues with the use of regular prostheses for this activity. The main goal of a cycling prosthesis is to efficiently transfer the power generated by the residual limb to the crank [4]. Therefore, a specially made prosthesis can be advantageous for cycling compared to a regular daily prosthesis.

The movement required for cycling involves the lower limbs and several muscle groups and joints, which interact with the bicycle to generate crank power; the points of interaction between the cyclist's body and the equipment include the saddle and handlebar [9]. The power generated by these muscle groups is divided throughout the crank and its phases. Distal muscle groups, such as the tibialis anterior, soleus, and gastrocnemius, provide balance for the ankle joint. Meanwhile, the quadriceps, rectus femoris, hamstrings, and gluteus maximus are responsible for power delivery during the power phase, with the quadriceps and gluteus maximus being the major power producers. The iliopsoas aids in lifting the leg during the recovery phase [4].

Unilateral transtibial amputations lead to the loss of the ankle joint and distal muscle groups, causing a decline of 15% of the torque generated by the ankle and the muscles used for stabilization (tibialis anterior, soleus, and gastrocnemius) [10]. This results in kinetic and kinematic asymmetries being the main issue among unilateral amputee cyclists [4,11]. Studies comparing cycling between individuals with unilateral transtibial prostheses and individuals without any prosthesis have shown that these asymmetries consist of several factors, including different ranges of movement (ROMs) in the knee and hip joints, leading to lower extensor moments. Muscle activation patterns differ between intact and amputated legs in individuals with unilateral amputation while cycling with prostheses. Previous research has shown that unilateral amputee cyclists delayed the peak muscle activity of the sound leg, reaching maximum activation later in the cycle compared to non-amputee cyclists [12]. Another factor contributing to asymmetry is the ability to deliver power through the prosthesis. Regular walking prostheses have moving parts that dissipate the energy delivered to the pedal. However, with the loss of distal muscle groups, this power cannot be generated [10,13]. The limb wearing a prosthesis in individuals with unilateral amputation presents considerably increased work asymmetry, at 24.5 ± 10.0% [11].

Amputee cyclists at the professional or competitive levels use specially made prostheses. These prostheses are often customized, do not have any points of articulation or movement, and can be made using several methods and materials. Various models exist, as each cyclist has their own prosthesis fabricated specifically for their needs, with no definite manufacturing guidelines. For the purposes of this research, the most common designs of cycling prostheses were divided into two models (Figure 1):

- **Rigid (a):** Prostheses fabricated by aligning the residual limb with the pedal through a single stiff rod. These models are commonly used by enthusiasts and professional cyclists.
- **Rigid off-center (b):** Similar to the previous model, this prosthesis is stiff; however, it contacts the pedal at a misaligned position in relation to the residual limb, providing a point of attachment similar to that of the tip of the foot on the pedal. These models are commonly worn by Paralympians.

Considering the lack of standardization in the design of cycling prostheses and the unique needs of individuals with amputation, the main objectives of this research are as follows:

1. To compare the efficiency of cycling using a unilateral transtibial cycling prosthesis with regular intact cycling.
2. To assess possible prosthesis models and their applications through comparisons between the attachment positions of the rigid and rigid off-center models.

To achieve the objectives listed above, biomechanical and surface electromyogram (sEMG) data were collected.

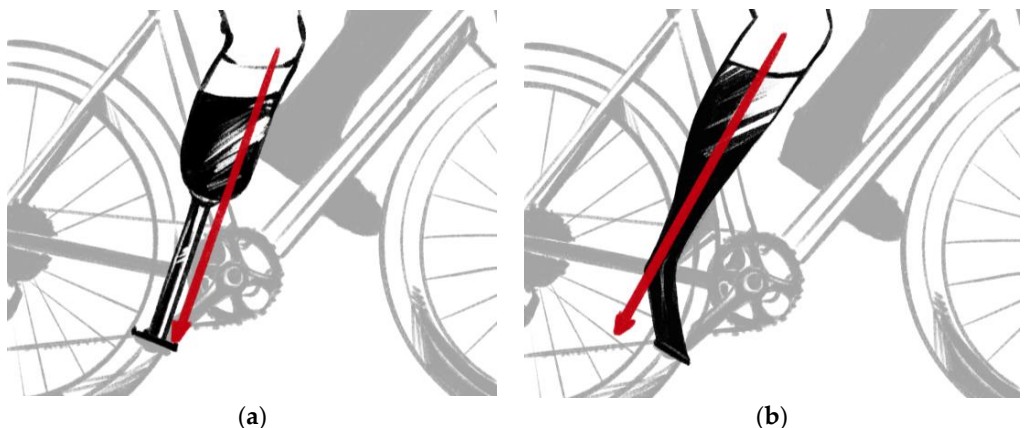

| (**a**) | (**b**) |

**Figure 1.** (**a**) Rigid prosthesis model and (**b**) rigid off-center prosthesis model. The red arrows indicate the alignment between the residual limb and the pedal.

## 2. Materials and Methods

### 2.1. Participants

Eight healthy able-bodied female (n = 5; age: 26.8 ± 4.7 years; height: 160.6 ± 2.7 cm; weight: 61.6 ± 4.7 kg) and male (n = 3; age: 21.6 ± 2.0 years; height: 173.3 ± 9.5 cm; weight: 66 ± 14.7 kg) participants were recruited. The inclusion criteria were age between 20 and 35 years and being familiar with the practice of cycling. Exclusion criteria encompassed presenting any sort of condition that could potentially be influenced or worsened by vigorous exercise, having severe discomfort during cycling, or having any musculoskeletal condition that could be aggravated by the tasks performed during the experiment.

All participants provided their written consent to participate in the study. They were asked to fill in a health questionnaire that enforced the exclusion criteria and answer a general questionnaire about cycling and leg dominance, as defined by the leg reaching the bottom dead position of the pedal first when commencing cycling. The majority of participants (n = 7) presented right-leg dominance. After the administration of the questionnaire, anthropometric data were collected and weight, height, and foot length (25 ± 1.2 cm) were measured; in addition, right leg inseam length (78 ± 4.1 cm) was determined, which was required to establish seat height.

### 2.2. Pre-Experiment Orthosis Fabrication

To simulate amputations below the knee, we employed custom-made orthoses (Arizono Orthopedic Supplies Co., Ltd., Kitakyushu, Japan) (Figure 2) attached below the knee on able-bodied participants.

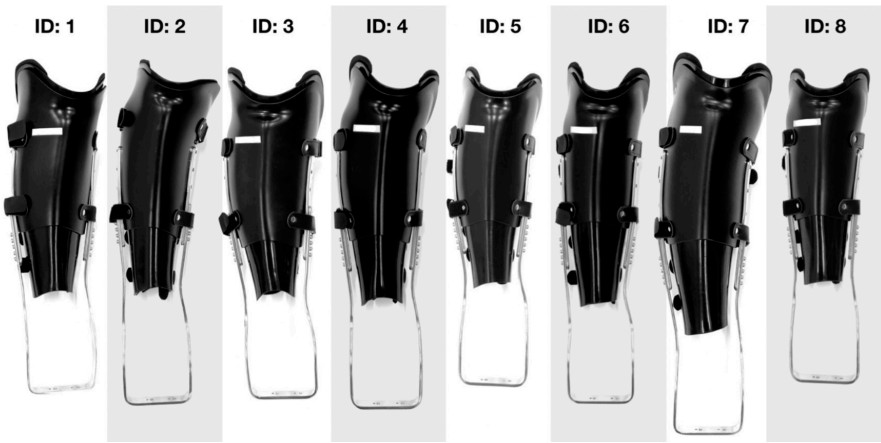

**Figure 2.** Orthosis models individually fabricated for corresponding participants.

The orthoses were fabricated by certified orthotists through an individual molding process, which required each participant to have their right leg measured for bony landmarks. A cast (Figure 3) was fabricated to create a mold, which was used for the modeling of an orthosis constructed using thermoplastics, aluminum, and fasteners (Figure 4). The size of the orthosis was determined by anatomical landmarks, with the highest point being the lateral epicondyle of the tibia and the lowest the lateral malleolus. All orthoses were fitted with a straight bar across the heel 2 cm away from the bottom of the participant's foot.

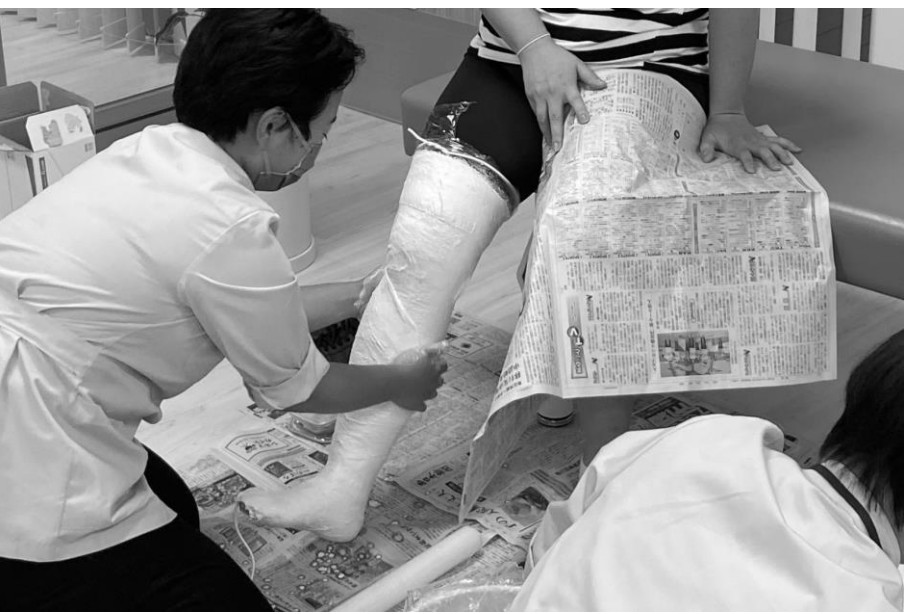

**Figure 3.** Cast fabrication process.

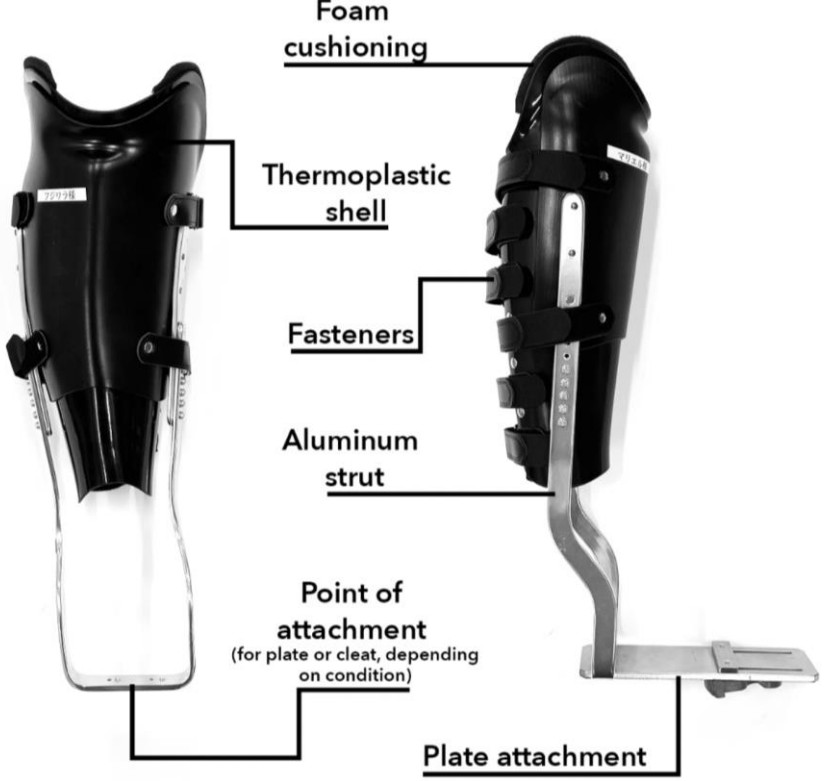

**Figure 4.** Orthosis parts.

These appliances provided the experimental setting to compare the data obtained through prosthesis-simulated conditions and regular cycling for the same individual. To simulate conditions similar to those experienced by cyclists wearing prostheses, aluminum struts coupled to a plate attachment provided the points of contact between the orthosis and the pedal. Therefore, the foot of the participant was prevented from contacting the pedal and the ankle joint was not employed in the cycling task.

### 2.3. Experimental Condition

The experiment was performed under three conditions, comprising the unilateral use of the orthosis on the right side with two different attachments points and regular cycling (Figure 5). All attachments involved a cleat (LOOK Keo, XPEDO Pedal Co., Culver City, CA, USA) for clip-on pedals.

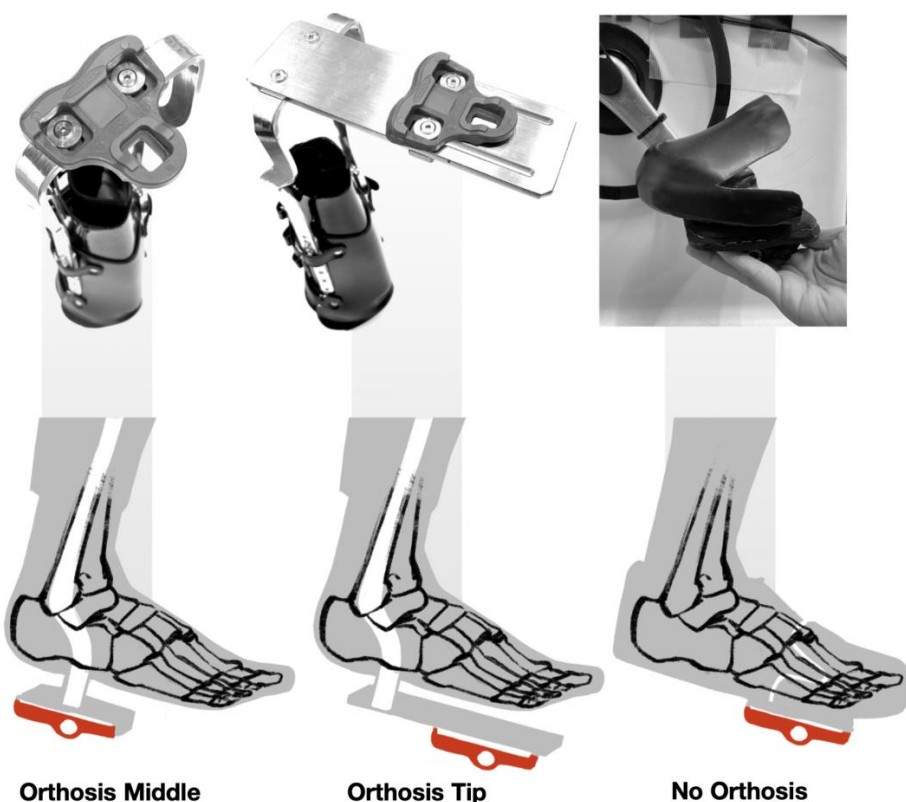

**Figure 5.** Attachments for orthosis at the middle and tip of the foot as well as with the regular cycling limiter.

### 2.3.1. Orthosis Middle (OM)

This condition simulated the rigid prosthesis model, with the lower leg attached to the pedal at the middle of the foot. The simulated effective prosthetic length was measured between the lateral epicondyle of the tibia, covered by the upper edge of the orthosis, and the attachment to the pedal, equivalent to the talus.

### 2.3.2. Orthosis Tip (OT)

This condition simulated the rigid off-center prosthesis model, using a plate to attach the tip of the foot to the pedal. The simulated effective prosthetic length was slightly longer than that in the OM condition and was measured between the upper edge of the orthosis and the plate attachment, which was placed at the level of the metatarsal phalangeal joints.

### 2.3.3. No Orthosis (NO)

This condition represented the regular cycling condition. It involves a limiter which secures the tip of the foot to the pedal at a predetermined position, similar to that in the OT condition, maintaining the pedal at the level of the metatarsal phalangeal joints.

### 2.4. Experimental Setup

Cycling Environment Setup

We assessed the cycling performance of the participants on an ergometer (AeroBike 75XL, Konami Sports Co., Ltd., Kanagawa, Japan). The seat height was determined using the Greg LeMond method (88.3% of the inseam length), and the handle height was altered according to each participant's preferences. The crank length was set at 170 mm, and regular factory-issued pedals were substituted with power meter pedals with clip-on fittings (Assioma Duo, Favero Electronics Srl., Arcade, Italy). All experiments were conducted in the Gymnasium at Ohashi Campus, Kyushu University.

For regular cycling, the clip-on pedals were adapted using 3D printed parts (Figure 6a) which simulated regular platform pedals. Thermoplastic fittings molded on the shoes used during the experiment were then bolted onto the platform pedals to provide the bracing of the foot position on the pedal. Clip-on pedals and cleats (Figure 6b) provided the attachment and ability to pull the pedals during the upstroke. Because the aim of this study was to evaluate the efficacy of recreational cycling, the fittings used on the pedals did not provide grip, but limited the extent of forward placement of the foot on the platform pedals. These fittings were present on both the right and left sides of the ergometer during the NO condition and on the left side during the two orthosis conditions.

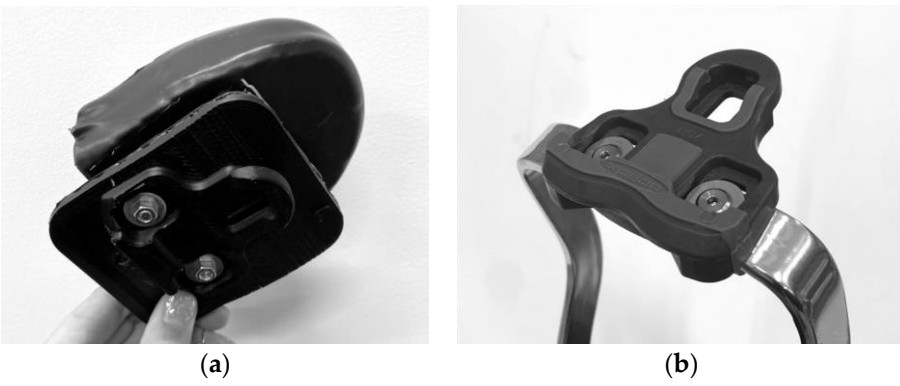

|       |       |
|:-----:|:-----:|
| (**a**) | (**b**) |

**Figure 6.** (**a**) 3D printed platform and its fitting compared to (**b**) a regular cycling cleat.

### 2.5. Experimental Protocol

Participants were asked to change into tight-fitting t-shirts and short pants of their preference, having different options at their disposal; however, they could also wear their own clothes if they fulfilled the requirements. For the cycling task, participants wore laceless training shoes (MW100, New Balance Athletics Inc., Boston, MA, USA) fitted according to their corresponding foot size. The shoes were worn on both feet during the NO condition and only on the left foot during orthosis conditions. Participants could wear their own socks if they fit snugly and covered the ankle.

During the experiment, participants took part in three trials lasting for 2 min each, with a 30 s break between trials and a 5 min break between conditions. Participants were allowed to wear the orthoses and practice during the 5 min break before each condition. The trials covered the six conditions randomly.

The participants performed cycling under 3 (foot position) × 2 (resistance) variables. The foot position was categorized as NO, OM, or OT (Figure 7). Each one of these conditions had two other variables, defined by a resistance setting on the ergometer: 40 W resistance, which simulates regular everyday cycling on a level path, and 60 W resistance, which replicates cycling up a slight incline. For 40 W, the participants were asked to maintain

a pace of 60 rpm, while the 60 W setting required a pace of 50 rpm. To maintain these cadences, the participants were aided by a metronome.

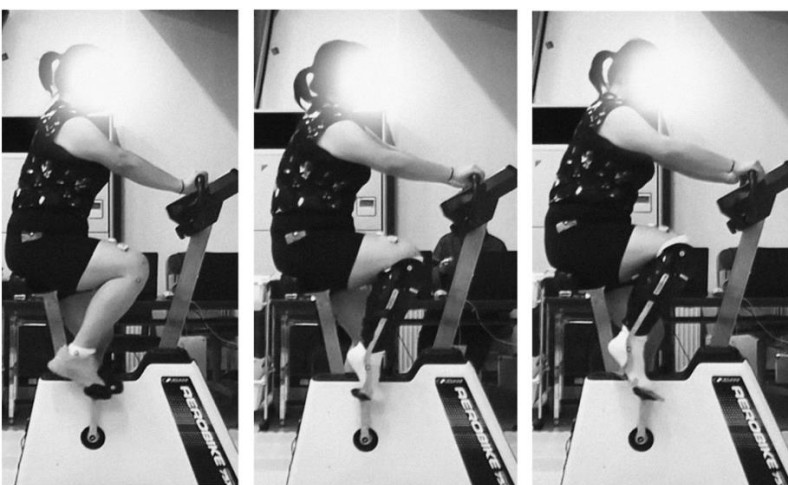

**Figure 7.** Cycling conditions with different attachments.

*2.6. Measurements*

2.6.1. Joint Angles

To assess the participants' cycling movement, a camera (HC-300M, Panasonic Co., Osaka, Japan) was placed on the right side of each participant to measure their joint angles and hip movement. The camera recorded the entire trial run for each condition at 59.94 frames/second and 1080 p resolution in the MTS format. To enable motion tracking, five color-contrasting auto-adhesive 16 mm markers were placed on the tight-fitting clothes of the participants over the following anatomical landmarks (Figure 8): right acromion, right greater trochanter, right lateral femoral epicondyle, right lateral malleolus, and right fifth metatarsal head.

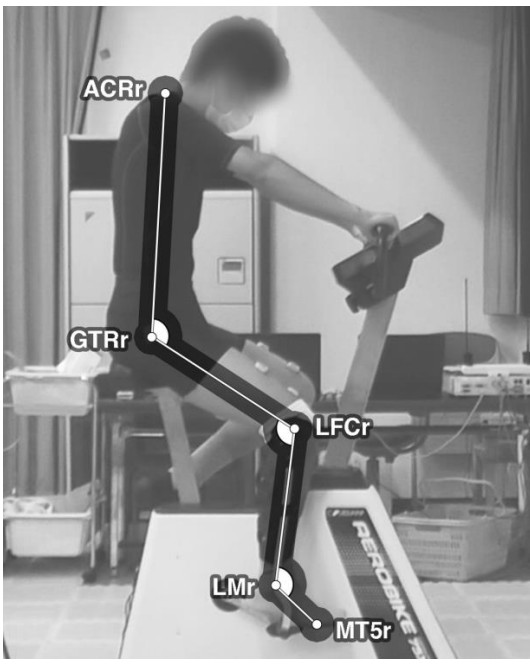

**Figure 8.** Markers were placed at the following points: ACRr, right acromion; GTRr, right greater trochanter; LFCr, right lateral femoral epicondyle; LMr, right lateral malleolus; and MT5r, right fifth metatarsal head. The image shows the points and measured angles.

For motion tracking, the resulting video data were processed using video editing software (Premiere Pro 15.4, Adobe Inc., San Jose, CA, USA). Joint angles were measured in every frame of the video by connecting the markers and creating vectors in OpenCV software. Vectors were drawn between the following markers: shoulder and hip, hip and knee, knee and ankle, and ankle and foot. Angles were calculated at the hip, knee, and ankle markers. Values for the angles between corresponding vectors were expressed in degrees.

### 2.6.2. EMG

For muscle activity measurements, six electrodes (WEB7000, NIHON Kohden Co., Tokyo, Japan) were placed on the proximal parts of the participants' lower limbs bilaterally to assess three muscles: semitendinosus, vastus lateralis, and vastus medialis. The electrodes were placed based on previous studies [9,14,15] by selecting the muscles that presented a more significant difference in activity during cycling between conditions. Specific placement points were defined according to SENIAM guidelines [16].

sEMG data were collected at 1000 Hz using a telemeter system. The signal was internally filtered at a bandwidth between 500 Hz and 15 Hz. For this experiment, sEMG data were sectioned into cycles using a Hall Sensor (Figure 9). The sensor was placed on the left side of the ergometer, with a 3D printed PLA (polylactic acid) wheel attached to the crank. The wheel comprised eight neodymium magnets that triggered the sensor at every 45 degrees, with a ninth magnet triggering a second sensor at the start of every cycle. Data for this sensor were collected along sEMG data through the telemeter, subsequently allowing data to be displayed and analyzed at different crank positions and cycling phases.

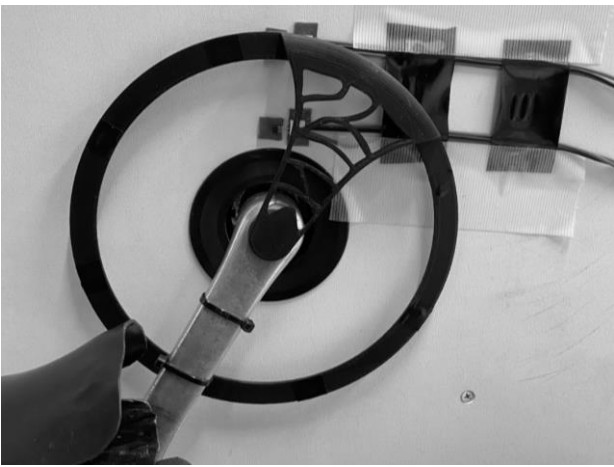

**Figure 9.** Hall Sensor.

### 2.6.3. Pedal Data

To measure different parameters of the power applied to the pedals, power meter pedals were installed on both sides of the ergometer. Data pertaining to the wireless pedal system were recorded at 1 Hz on GoldenCheetah ver. 3.5 software through an ANT+ receptor. The following parameters were measured:

- **Cadence and Power:** Cadence measures the number of crank revolutions per minute, calculated in rpm. Power measures the energy output applied to the pedals, calculated in watts (W). Instant power (P) can be either positive or negative, depending on the position of the crank. Its values used in the experiment were output as P average. Both parameters were measured only with the intent of monitoring target settings for each condition.
- **L/R Balance:** Balance measures the energy output by each leg contributing to one crank revolution. Because it was calculated as a percentage, and only values on the

right side were recorded, values higher or lower than 50 indicate greater or lesser asymmetry, respectively.

- **Left and right torque effectiveness:** This measures the power output delivered to the pedals. It was calculated as a percentage, with a value of 100 indicating that all the force applied by the foot was being translated as a force vector; hence, values lower than 100 indicate the delivery of effective force percentage. It was calculated as follows:

$$n = \frac{(P_+ + P_-)}{P_+}$$

- **Left and right pedal smoothness:** The smoothness of the application of power to the pedals; values closer to 100 indicate that power was delivered constantly across one revolution of the crank. The data were collected as percentages, and values were calculated as follows, with $P_{avg}$ indicating average power, and $P_{max}$ corresponding to maximum power:

$$n = \frac{P_{avg}}{P_{max}}$$

### 2.6.4. Perceived Exertion

Participants self-assessed their exertion using a Borg 14-category scale with scores ranging 6–20 [17]. The participants were prompted to report a number on the scale 15 s before the end of each trial.

### 2.7. Statistical Analysis

Statistical analysis was conducted in SPSS software (version 21.0, IBM Co., Armonk, NY, USA). Two-way repeated variance analysis (two-way ANOVA) was performed to compare mean values for each parameter and for all participants (n = 8). Factors for sEMG were established as 2 (position) × 2 (resistance). sEMG data for the NO position were used to normalize values for OT and OM, as these data were presented in % over NO. This method was chosen instead of normalization using maximum voluntary contraction in order to prevent over-exertion of the participants during trials while considering the main objective of comparison, namely, assessing differences between NO and orthosis conditions. Other data, including values of the joint angles (hip, knee, and ankle), hip height (left and right), pedal values (left and right balance, left and right torque effectiveness, left and right pedal smoothness), and perceived exertion (Borg scale), were analyzed with 3 (position) × 2 (resistance) factors.

Bonferroni-corrected pairwise comparisons were conducted to examine the main differences between positions or resistance settings as well as the interaction between both. The significance level ($p$-value) for this study was 5%, and all results are presented as mean ± standard deviation except otherwise stated.

### 3. Results

#### 3.1. Joint Angles and ROM

Figures 10 and 11 respectively demonstrate the mean angles measured at the hip and knee joints at different foot positions and two workloads. For the hip and knee joints, the ANOVA results showed the main effect of position (hip: F (1,7) = 24.623, $p < 0.01$, $\eta^2 = 0.779$, knee: F (1,7) = 81.725, $p < 0.01$, $\eta^2 = 0.921$). No significant interactions were found between position and workload conditions. Post hoc analyses indicated that angles with OT were significantly smaller than angles with NO and OM (OT and OM = hip: $p = 0.01$, knee: $p < 0.01$, OT and NO = hip: $p = 0.01$, knee: $p < 0.01$), making this the condition with the most acute angles.

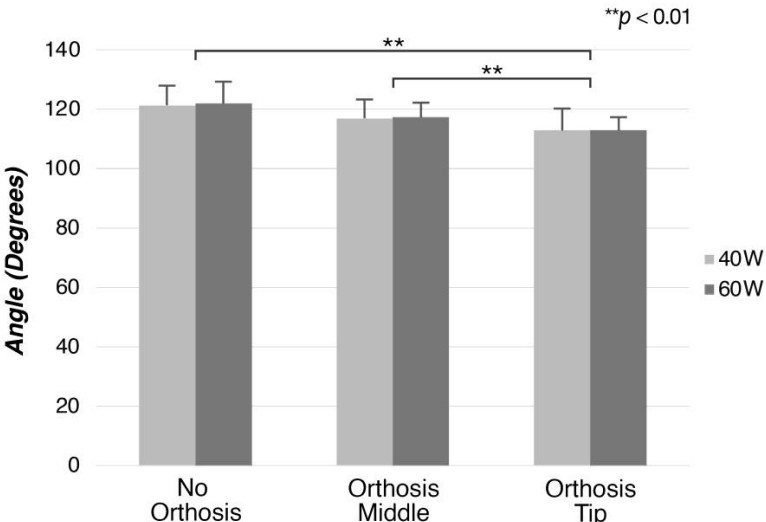

**Figure 10.** Mean hip angles for the three orthosis conditions and two workloads.

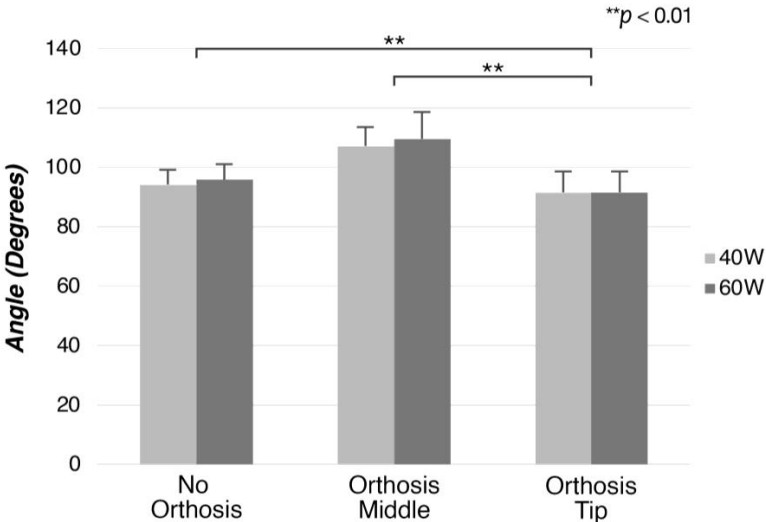

**Figure 11.** Mean knee angles for the three orthosis conditions and two workloads.

### 3.2. Muscle Activity

Mean sEMG data were plotted on a circular graph. Data were calculated as the % variation over the NO condition, which is represented by the darker circle at the center of each graph. The angles in the graphs correlate with the crank position throughout the cycle, thereby depicting the mean muscle activity for different crank positions. Reading the graphs clock-wise, the power phase, which corresponds to the phase during which force is applied to the pedal, is shown between $0°$ and $180°$, followed by the recovery phase between $180°$ and $360°$.

For the leg wearing the orthosis, the semitendinosus muscle (Figure 12) demonstrated increased muscle activity (14~18%). The sound leg showed a slight peak in muscle activity under all resistance conditions at the beginning of the recovery phase while presenting minimum to no change in overall muscle activation (0~2%). The leg wearing the orthosis showed a noticeable overall increase in activity over NO (14~18%) and a very pronounced shift in peak activation between orthosis conditions. The OM conditions demonstrated a peak starting at the end of the recovery phase until roughly halfway through the power phase, while the OT conditions showed a peak starting at the end of the power phase and throughout the recovery phase.

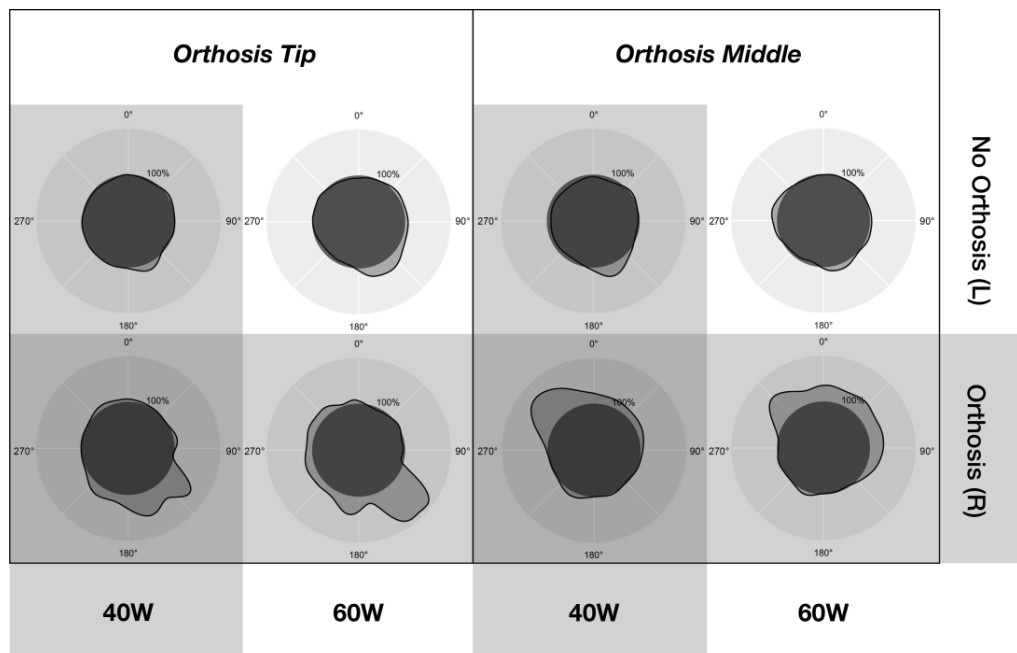

**Figure 12.** Mean semitendinosus muscle activity in relation to crank angle. The NO condition values are denoted by the darker area in the center.

The vastus medialis (Figure 13) showed overall negative mean values for the affected leg. Muscle activation increased in the unaffected leg (9~22%), especially under the 40 W resistance OT condition (16%). This marked increase started at the beginning of the power phase, lasting up until 135°. Meanwhile, the leg wearing the orthosis showed a decrease in muscle activation, with OM 60 W showing the largest decrease (−24%), again starting at 135° and lasting throughout the recovery phase until the switch to power phase, with a distinguishable peak at 315° across all plotted conditions.

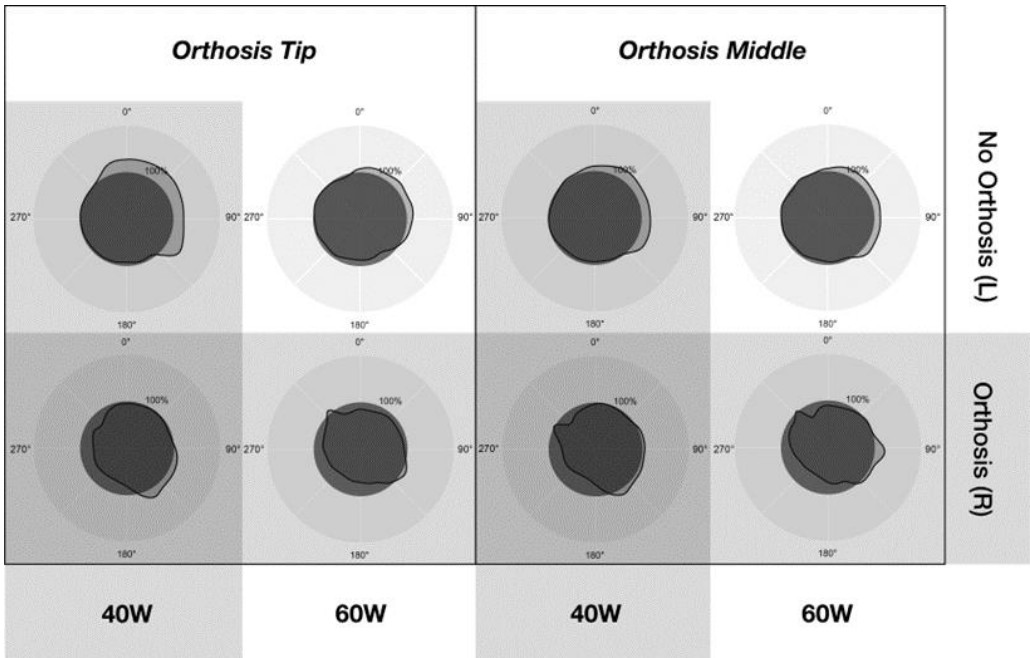

**Figure 13.** Mean vastus medialis muscle activity in relation to crank angle. The NO condition values are denoted by the darker area in the center.

The vastus lateralis (Figure 14) presented a similar trend, with overall negative values for the leg wearing the orthosis and an increase in muscle activation (5~12%) on the unaffected leg starting at the beginning of the power phase and until 135°, with 40 W resistance in the OT position showing a greater increase across all conditions. A decrease (−13~19%) in muscle activation for all conditions in the leg wearing the orthosis was observed during the recovery phase, with both OM conditions showing the largest decrease (−19%).

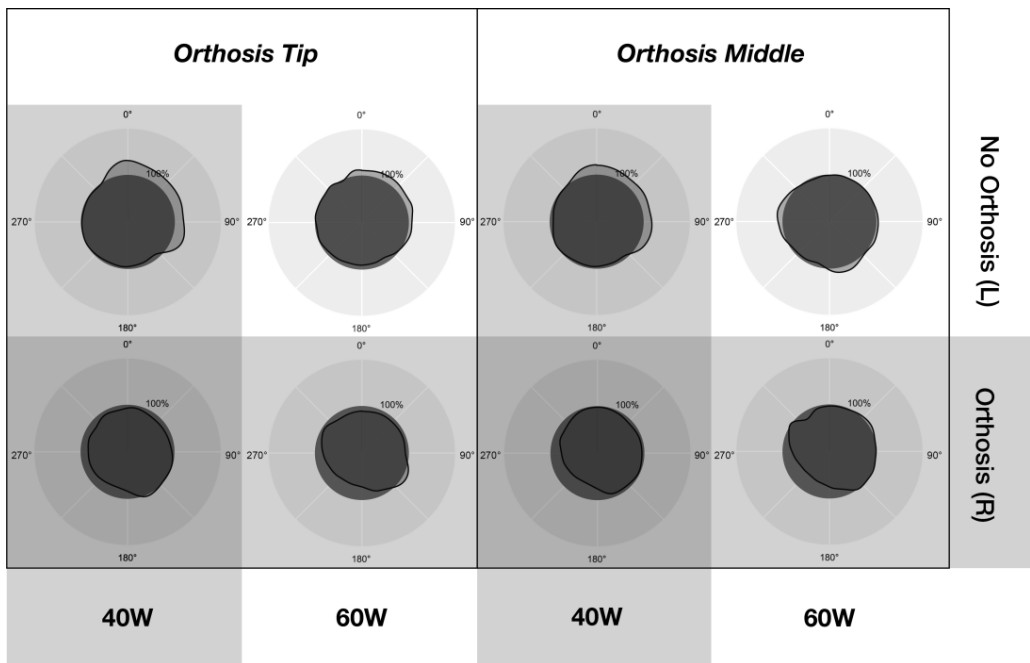

**Figure 14.** Mean vastus lateralis muscle activity in relation to crank angle. The NO condition values are denoted by the darker area in the center.

### 3.3. Pedal Data

Mean left and right balance scores are presented in Figure 15. The main effect of position (F (1,7) = 31.382, $p < 0.01$, $\eta^2$ = 0.818) and workload (F (1,7) = 45.087, $p < 0.01$, $\eta^2$ = 0.866) was demonstrated using ANOVA, with both orthosis conditions showing significantly lower means than NO, as evident through the results of post hoc analyses (NO and OM: $p = 0.01$, NO and OT: $p < 0.01$). No significant interaction was found between position and workload.

The mean torque effectiveness is shown in Figure 16a for the unaffected leg and Figure 16b for the leg wearing the orthosis. The results of ANOVA for both sides showed the main effect of position (Left: F (1,7) = 19.399, $p < 0.01$, $\eta^2$ = 0.735, Right: F (1,7) = 20.939, $p < 0.01$, $\eta^2$ = 0.749) and workload (Left: F (1,7) = 624.530, $p < 0.01$, $\eta^2$ = 0.989, Right: F (1,7) = 799.807, $p < 0.01$, $\eta^2$ = 0.991). Post hoc analyses revealed that for the sound leg, the results of both orthosis conditions were significantly different from those of NO (NO and OM: $p = 0.01$, NO and OT: $p < 0.01$), while for the affected leg the values for OT were significantly different from those for the other two conditions (OT and NO: $p = 0.01$, OT and OM: $p = 0.01$). Hence, OT showed the lowest percentage of power being delivered to the pedal. The interaction between position and workload yielded no significant effects.

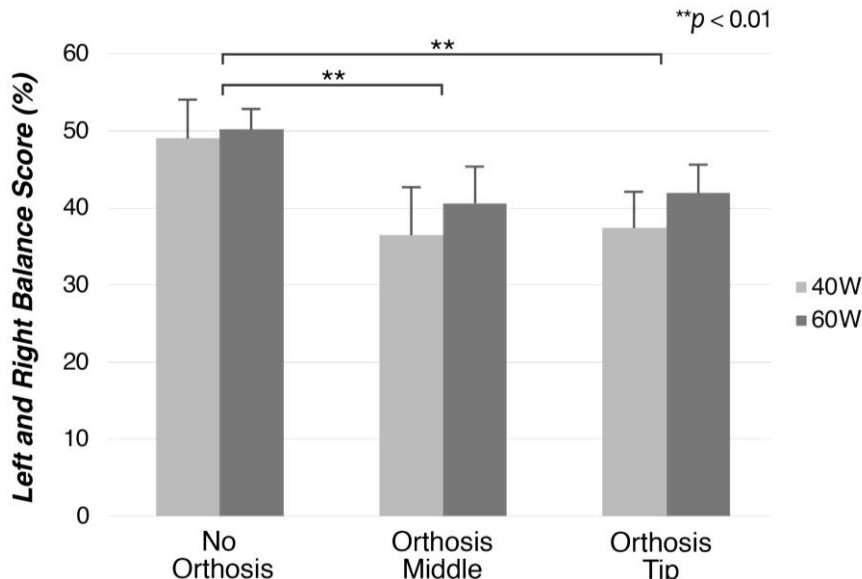

**Figure 15.** Left and right balance scores implying percentage of power production by the right (orthosis) foot for the three orthosis conditions and two workloads.

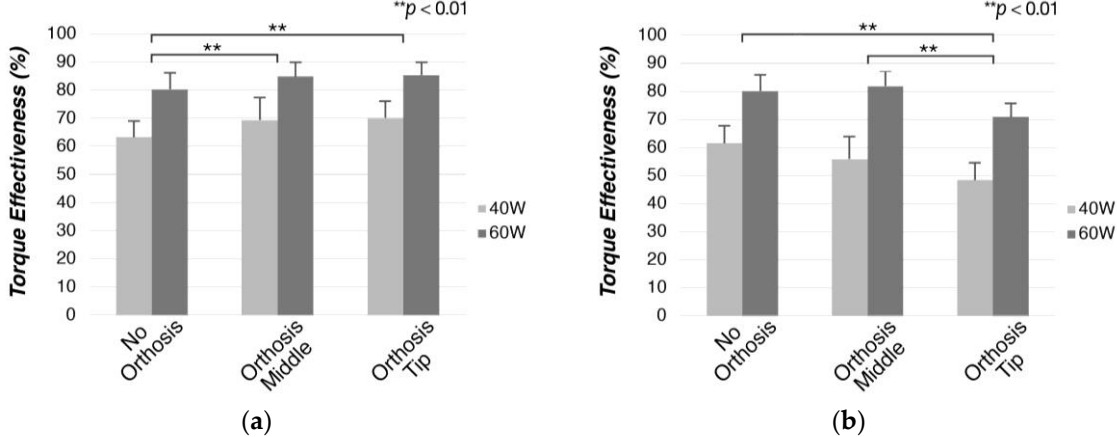

**Figure 16.** (**a**) Mean torque effectiveness for the unaffected leg under the three orthosis conditions and two workloads and (**b**) mean torque effectiveness for the affected leg under the three orthosis conditions and two workloads.

The mean pedal smoothness in the sound leg and the leg wearing the orthosis is shown in Figure 17a,b, respectively. Again, the main effects of position (Left: F $(1,7)$ = 7.352, $p < 0.01$, $\eta^2$ = 0.512, Right: F $(1,7)$ = 14.630, $p < 0.01$, $\eta^2$ = 0.676) and workload (Left: F $(1,7)$ = 188.884, $p < 0.01$, $\eta^2$ = 0.964, Right: F $(1,7)$ = 383.631, $p < 0.01$, $\eta^2$ = 0.982) were evident according to the ANOVA results and the interaction between position and workload was significant for the right leg (F $(2,14)$ = 4.185, $p < 0.05$, $\eta^2$ = 0.374). For the left leg, OT showed the highest smoothness percentages (NO and OT: $p < 0.01$), while for the right leg this position demonstrated the lowest percentages (OM and OT: $p < 0.01$, NO and OT: $p < 0.05$).

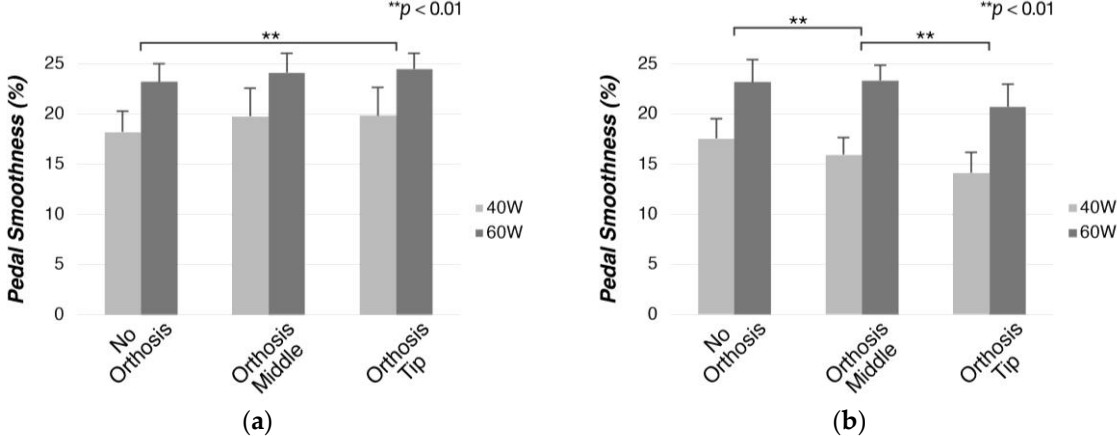

(**a**)　　　　　　　　　　　　　　(**b**)

**Figure 17.** (**a**) Mean pedal smoothness for the unaffected leg under the three orthosis conditions and two workloads and (**b**) mean pedal smoothness for the affected leg under the three orthosis conditions and two workloads.

*3.4. Perceived Exertion:*

The perceived exertion results are shown in Figure 18. The ANOVA results showed the main effect of workload only (F (1,7) = 30.097, $p < 0.01$, $\eta^2 = 0.811$). No significant statistical interaction was found between pedal placement conditions or resistance.

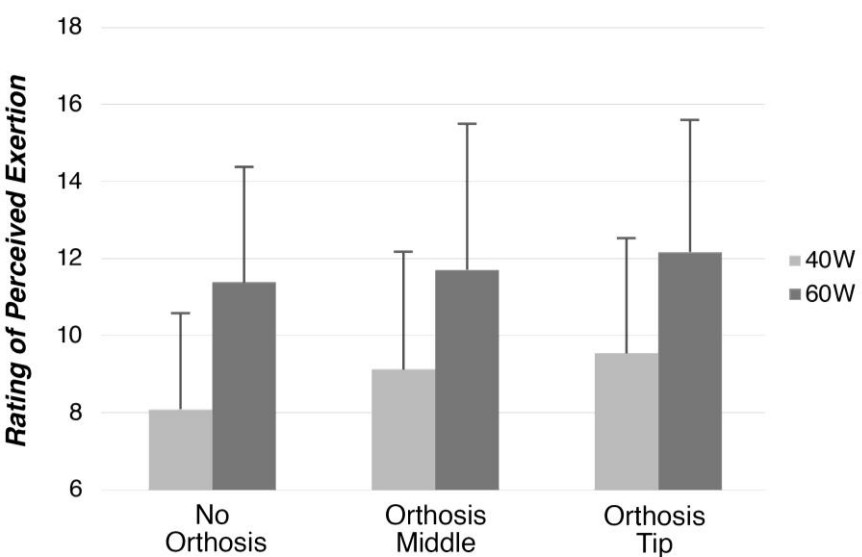

**Figure 18.** Mean perceived exertion for three orthosis conditions and two workloads.

## 4. Discussion

In this study, we aimed to explore the efficiency of transtibial amputee cycling for recreation. The study proposes a new approach for quantifying, classifying, and evaluating cycling practice using orthoses that simulate cycling conditions similar to those of prostheses on non-amputee participants. This allowed for the comparison of data collected in an intact cycling condition (NO) and two simulated prosthesis conditions (OM/OT) performed by the same participant. Overall, the results showed increased asymmetries in all measured parameters and different points of attachment of the orthosis on the pedal, modifying the levels of power delivery, kinematic asymmetries, and muscle activity patterns at different intensities.

The presented asymmetries were grouped by coincidental factors. The results of certain parameters differed similarly between orthosis attachment positions and normal cycling; therefore, they were classified as asymmetries due to cycling under simulated

prosthesis conditions. Other results showed clear differences between the attachment positions; hence, this factor was considered as the cause of the differences. We discuss these results separately below.

### 4.1. Asymmetries Due to Cycling with a Prosthesis

Both simulated prosthesis conditions showed joint movement results that were similarly different from intact cycling. This was reflected in the higher hip flexion (Figure 10). Considering that cycling prostheses, as simulated by the orthoses in this study, are stiff and do not dorsiflex or plantarflex, the length between the upper point of attachment of the orthosis to the leg and the pedal remains constant throughout the cycle [12,15]. Consequently, the hip joint has to flex more towards the top of the cycle to allow for movement completion.

During the power phase (0° to 180°), the knee extensor muscles (the vastus lateralis and vastus medialis) exert power to push the pedal downward. During this phase, in both orthosis conditions the leg wearing the orthosis showed a decrease in sEMG activity below NO levels, whereas the sound leg showed an increase in muscle activity. Previous findings [2,12,14] have shown similar activation patterns to those presented in this study for amputee cyclists, while showing increased muscle activity in the proximal portion of the prosthetic leg, which contradicts the findings of this study. This discrepancy can be attributed to the difference in cycling intensity reported among studies; resistance (140 to 200 W) and cadence levels (70 to 90 rpm) were higher in previous studies than those in the present study. Therefore, it can be postulated that the participants in this study selected a unique strategy for cycling, choosing to favor the sound leg at resistance levels that allowed for it.

Pedal data, including left–right balance, torque effectiveness, and pedal smoothness, were influenced by both workload and orthosis setting. A higher workload (60 W) showed a better left–right balance (Figure 15), with torque effectiveness (Figure 16) and pedal smoothness (Figure 17) values for the leg wearing the orthosis being closer to those for NO, thereby, enabling better symmetry. Research in professional-level amputee cycling [14,18] has shown an increase in kinematic symmetry at higher workloads. Furthermore, previous findings have noted enhanced power delivery at higher resistances in regular cycling [19–21] because of the workload requirements. For the unaffected side, a slight increase in force parameters for the orthosis conditions over NO was observed (Figures 15–17), which could be attributed to unilateral amputee cyclists employing their sound leg more at lower cadences [12,13]; these results are in line with sEMG results.

The asymmetries observed in this study can be defined as a unilateral difficulty in exerting and directing forces [13]. The use of the orthoses caused an increase in muscle activity on the unaffected side and presented more symmetrical power delivery at the higher resistance setting, indicating the use of a unique pedaling technique with the orthosis employing the sound leg more. This technique enables better kinetic symmetry at a higher resistance level. Factors such as difficulty in completing cycles while wearing the orthoses due to differences in joint coordination among joints in the leg, an immobilized ankle joint, and unfamiliarity in pedaling with orthoses could have led the participants to use their sound leg more (at the expense of symmetry) at lower workloads, while a higher output from the orthosis side was not necessary. Supporting this, no significant differences were found in the perceived rate of exertion (Figure 18) between intact cycling and the simulated prosthesis condition. Experienced unilateral amputee cyclists aim for more symmetrical power delivery, often in exchange for kinematic symmetry [11,15], contradicting the findings in this study. This is owing to better power output symmetry at higher workloads and cadences leading to lower metabolic costs, which affects endurance during the extended practice of cycling [22–26]. Elite amputee cyclists present familiarization with cycling practice and the use of prostheses at competitive levels, and use developed motor strategies and pedaling techniques that enable enhanced power output symmetry [12,14,15,27].

*4.2. Effects of Pedal Attachment Position*

The main factor affecting joint movement was the angle of alignment between the knee and the pedal attachment. The misalignment between the knee and the pedal was higher in the OT position, resulting in higher knee flexion (91.5°~91.6°), with the leg extending less throughout the cycle [4]. Maintaining the alignment closer to 90° (OM position) results in higher extension at the hip (116.9°~117.3°), with the leg extending further down. Regarding pedal data, the leg wearing the orthosis showed higher pedal smoothness and torque effectiveness in the OM position, both being significantly different from those in OT and closer to those in regular cycling (Figures 16 and 17). This implies better power delivery to the pedals and ease of movement in the OM condition. Therefore, it can be postulated that power delivery is affected more by higher knee flexion (OT) than higher hip extension (OM). Although the hip and knee joints generally contribute equally to the cycling movement [28,29], at recreational levels cyclists produce more power at the knee than at the hip due to the pedaling technique employed [30,31].

Between both orthosis conditions, the semitendinosus muscle showed a higher degree of activation later in the cycle than the NO level activation pattern. This was the only monitored muscle with two biomechanical roles, i.e., hip extensor and knee flexor [32]. The muscle showed higher activity between 110° and 250° for OT and between 300° and 100° for OM (Figure 12). Normally [33,34], the semitendinosus muscle is most active at the top dead center of the crank or at the beginning of the cycle as a hip extensor. Although demonstrating a peak activity that commenced earlier than previously reported, in the OM position the muscle showed a pattern consistent with being a hip extensor. However, for the OT condition, the pattern was compatible with the knee flexor role [32]. This indicates a possible change in the biomechanical role for this orthosis position. While joint movement data show that the knee flexes more under OT, this could also suggest that the semitendinosus compensates for the lower levels of activation in the knee extensor muscles during the power phase.

Previous research in regular ergometer cycling has shown that changes in pedal attachment do not significantly affect parameters such as muscle activity [35], pedal forces [36], and knee joint movement [37]. However, the data presented here demonstrate that the change in attachment position under this study's conditions can affect all of these factors. This highlights the role of the ankle joint in cycling, which compensates for the constrained angle between the pedal and the foot attachment; in addition, it enables the production of power and aids in directing the power vector [38–40]. Therefore, it can be deduced that the main factors contributing to the observed differences between the attachment positions are the lack of ankle movement and the angle of alignment between the pedal and the knee.

In conclusion, the implications for both orthosis positions and their respective representative prosthesis models can be summarized as follows:

- **Orthosis Middle:** Presents similar muscle activation levels in both resistance settings, and presents power delivery to the pedals that is closer to levels with no orthosis. Therefore, this type of prosthesis can be prescribed to individuals with amputation in the initial stages of rehabilitation, allowing them to gradually increase their workload while maintaining smoother power delivery and more consistent muscle activation. Patients with transtibial amputation have a greater risk of developing hip osteoarthritis [41,42], in addition to potential hip asymmetries. Therefore, this prosthesis model could be recommended under these factors, as it presents levels closer to no orthosis for hip joint movement.

- **Orthosis Tip:** Presents more asymmetric power delivery at lower workloads and cadences. Patients who might have special requirements for the knee joint could benefit from this prosthesis model, as that for this joint the results were closer to NO levels. Overall, because of muscle activation patterns observed under both workloads, orthosis tip shows more asymmetry at lower workloads, being closer to no orthosis levels at higher resistance settings. This could indicate more ideal application under

conditions similar to amateur or professional cycling, which has been indicated in previous studies [2,4].

### 4.3. Limitations

The limitations of this study include the use of orthoses to simulate limb loss. While orthoses can be an effective way of simulating and standardizing the experimental conditions, many factors directly related to amputation and how the muscles adapt cannot be emulated using this setting. Muscles often change their biomechanical role after an amputation [9,12], which can be measured directly only by performing experiments with individuals with amputations. The use of a prosthesis after amputation leads to the adaptation to cycling, and consequently to the development of a cycling technique. As highlighted by the asymmetries seen in the orthosis conditions, this technique, which is developed through the continuous practice of cycling [31,43], often influences the biomechanics of prosthesis cycling. In our setting, participants were exposed to cycling with the orthosis for a limited duration of time and at light workload conditions; thus, the same adaptation to cycling seen in individuals with prosthesis did not occur.

Several factors related to the use of the bicycle with the limitations of the orthosis could not be assessed using an ergometer. The participants were not required to maintain the bicycle in a balanced state; furthermore, tasks involving mounting the bicycle, starting the cycling motion, and the attachment of the simulated prosthetic foot on the pedal were not appraised. Moreover, the use of a power meter pedal system, which could only output mean power, limited the scope for comparison with previous studies that demonstrated a switch in power peaks [14]. Lastly, seat height was determined using the Greg LeMond method, which was chosen as a standardization tool; this method is mostly used in professional cycling contexts, and recreational cyclists might choose a different seat height depending more on comfort [44].

### 4.4. Conclusions

In conclusion, simulated prosthesis conditions showed better symmetry between the unaffected leg and the leg wearing the orthosis at higher workloads. Furthermore, participants used a technique that employed the sound leg more at lower resistance conditions. Regarding the different attachment positions, OM showed power delivery closer to NO levels, while OT showed lower and more asymmetric levels of power delivery. Moreover, the semitendinosus muscle changed its biomechanical role for this position. These results suggest that better symmetry for unilateral transtibial amputees practicing cycling at recreational levels can be achieved with the OM attachment position and at higher workloads.

**Author Contributions:** Conceptualization, H.S.F., W.L.Y., K.M. and S.M.; methodology, H.S.F., W.L.Y. and P.Y.L.; software, W.L.Y.; validation, K.M.; formal analysis, H.S.F.; investigation, H.S.F.; resources, P.Y.L., K.M. and S.M.; data curation, W.L.Y. and S.M.; writing—original draft preparation, H.S.F.; writing—review and editing, S.M.; visualization, H.S.F. and W.L.Y.; supervision, W.L.Y., P.Y.L. and S.M.; project administration, S.M.; funding acquisition, S.M. All authors have read and agreed to the published version of the manuscript.

**Funding:** This work was supported by the Tateisi Science and Technology Foundation 2021 Research Grant (A), Grant Number 2211026.

**Institutional Review Board Statement:** This study was approved by the Ethics Committee of the Faculty of Design at Kyushu University (approval number 389, 10 November 2020).

**Informed Consent Statement:** Informed consent was obtained from all participants in the study.

**Data Availability Statement:** The data that support the findings of this study are available from the corresponding author upon reasonable request.

**Acknowledgments:** We would like to thank Naoki Chizawa and Takahito Oda of Arizono Orthopedic Supplies Co., Ltd., Kitakyushu, Japan, and Teppei Morimoto, CPO and CEO of JST Fit Inc., for their cooperation in fabricating the experimental orthoses.

**Conflicts of Interest:** The authors declare no conflict of interest.

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
