# Peer review of "Biomechanical Analysis of Recreational Cycling with Unilateral Transtibial Prostheses"

_prosthesis, doi:10.3390/prosthesis5030052_

Round 1

Reviewer 1 Report

Dear Authors,

thank you for your contribution. Great work! I only have minor revisions

Introduction:

Line 52: there are several muscles involved and joints, not only those. Please be more general.

Methods:

Please put the patient demographic information at the beginning of the results and add a table reporting them.

Discussion:

Please discuss further your limitations

English level is ok

Reviewer 2 Report

The article is interesting and well-prepared. I have no comment, but the Authors could show the differences between the unaffected side for all cases and levels of resistance.

This would show what differences occur for the leg NO and which are closer to the values for the unaffected side.

Despite this, the article is suitable for publication in its present form.
